# Identification of cell type specific ACE2 modifiers by CRISPR screening

**Emily J. Sherman**[1], **Carmen Mirabelli**[2], **Vi T. Tang**[3,4], **Taslima G. Khan**[4,5], **Kyle Leix**[1], **Andrew A. Kennedy**[6], **Sarah E. Graham**[7], **Cristen J. Willer**[7,8,9], **Andrew W. Tai**[2,6,10], **Jonathan Z. Sexton**[6,11], **Christiane E. Wobus**[2], **Brian T. Emmer**[1]*

1 Department of Internal Medicine, Division of Hospital Medicine, University of Michigan, Ann Arbor, Michigan, United States of America, 2 Department of Microbiology and Immunology, University of Michigan, Ann Arbor, Michigan, United States of America, 3 Department of Molecular and Integrative Physiology, University of Michigan, Ann Arbor, Michigan, United States of America, 4 Life Sciences Institute, University of Michigan, Ann Arbor, Michigan, United States of America, 5 Chemical Biology Program, University of Michigan, Ann Arbor, Michigan, United States of America, 6 Department of Internal Medicine, Division of Gastroenterology and Hepatology, University of Michigan, Ann Arbor, Michigan, United States of America, 7 Department of Internal Medicine, Division of Cardiovascular Medicine, University of Michigan, Ann Arbor, Michigan, United States of America, 8 Department of Computational Medicine and Bioinformatics, University of Michigan, Ann Arbor, Michigan, United States of America, 9 Department of Human Genetics, University of Michigan, Ann Arbor, Michigan, United States of America, 10 VA Ann Arbor Healthcare System, Ann Arbor, Michigan, United States of America, 11 Department of Medicinal Chemistry, College of Pharmacy, University of Michigan, Ann Arbor, Michigan, United States of America

* bemmer@med.umich.edu

## Abstract

SARS-CoV-2 infection is initiated by binding of the viral spike protein to its receptor, ACE2, on the surface of host cells. ACE2 expression is heterogeneous both *in vivo* and in immortalized cell lines, but the molecular pathways that govern ACE2 expression remain unclear. We now report high-throughput CRISPR screens for functional modifiers of ACE2 surface abundance. In liver-derived HuH7 cells, we identified 35 genes whose disruption was associated with a change in the surface abundance of ACE2. Enriched among these ACE2 regulators were established transcription factors, epigenetic regulators, and functional networks. We further characterized individual HuH7 cell lines with disruption of *SMAD4*, *EP300*, *PIAS1*, or *BAMBI* and found these genes to regulate ACE2 at the mRNA level and to influence cellular susceptibility to SARS-CoV-2 infection. Orthogonal screening of lung-derived Calu-3 cells revealed a distinct set of ACE2 modifiers comprised of *ACE2*, *KDM6A*, *MOGS*, *GPAA1*, and *UGP2*. Collectively, our findings clarify the host factors involved in SARS-CoV-2 entry, highlight the cell type specificity of ACE2 regulatory networks, and suggest potential targets for therapeutic development.

## Author summary

The amount of ACE2 on the surface of human cells is an important determinant of SARS-CoV-2 infection, but the molecular pathways that regulate ACE2 remain poorly understood. Identification of these pathways may clarify host factors involved in COVID-19

**Data Availability Statement:** All relevant data are within the manuscript and its Supporting information files.

**Funding:** This research was supported by the National Institutes of Health National Heart, Lung,

and Blood Institute (BTE, K08-HL148552),
University of Michigan Frankel Cardiovascular
Center (BTE), A. Alfred Taubman Medical Research
Institute (BTE), Michigan Institute for Clinical and
Health Research (CM), Marie-Sklodowska-Curie
Global Actions Fellowship (CM, GA-841247), and
University of Michigan Biological Scholars
Program (CEW). The funders had no role in study
design, data collection and analysis, decision to
publish, or preparation of the manuscript.

**Competing interests:** The authors have declared
that no competing interests exist.

outcomes and offer targets for therapeutic development. ACE2-targeted therapies may
furthermore be less susceptible than viral spike-targeted therapies to evasion by SARS-
CoV-2 variants. To systematically identify regulators of human ACE2, we therefore per-
formed high-throughput CRISPR screening for modifiers of ACE2 surface abundance in
HuH7 liver-derived and Calu-3 lung-derived cell lines. Unexpectedly, aside from *ACE2*
itself, we identified distinct sets of ACE2 modifiers in either cell line. For a subset of ACE2
regulators, we validated their functional effect on ACE2, confirmed their relevance to
SARS-CoV-2 infection, and clarified their level of regulation. Our findings demonstrate
the important influence of cell type on investigations of SARS-CoV-2 infection and nomi-
nate candidate pathways for ACE2-targeted therapeutic development.

## Introduction

ACE2 plays a critical role in SARS-CoV-2 infection by serving as the cellular receptor for viral
entry [1]. Inhibition of endogenous ACE2 disrupts SARS-CoV-2 entry into permissive cell
lines while heterologous expression of ACE2 in non-permissive cell lines renders them suscep-
tible to infection [2–4]. Transgenic expression of human ACE2 sensitizes mice to SARS-CoV-
2 infection with recapitulation of pathologic hallmarks of COVID-19 [5–7].

Given its importance in SARS-CoV-2 infection, the interaction between the viral spike pro-
tein and ACE2 is an attractive target for therapeutic development. Vaccines against the spike
protein are broadly efficacious in reducing the number and severity of COVID-19 infections
[8], but the rapid evolution of SARS-CoV-2 raises concern for the potential of SARS-CoV-2
variants to escape immunity induced by either vaccines or prior infection [9–12]. As an alter-
native strategy, disruption of host ACE2 may similarly prevent SARS-CoV-2 infection in a
manner that is less susceptible to viral evolution. ACE2-targeted therapies may have broader
clinical applications, as ACE2 also serves as the cellular receptor for other respiratory viruses
such as SARS-CoV-1 [13] and HCoV-NL63 [14]. Additionally, ACE2 is an important physio-
logic regulator of the renin-angiotensin and kallikrein-kinin systems, and its dysregulation has
been implicated in pulmonary and systemic hypertension, cardiac fibrosis, atherosclerosis,
and acute respiratory distress syndrome [15]. However, no ACE2-targeted therapies have been
clinically approved and their development is limited by uncertainty in the molecular pathways
that regulate ACE2.

Since the onset of the COVID-19 global pandemic, several studies have applied single-cell
RNA-seq to examine *ACE2* expression in tissues of humans and animal models [16–19]. These
studies have consistently found *ACE2* mRNA expression to be heterogeneous among cell types
within a given tissue. *ACE2*-expressing cells include alveolar type 2 cells in the lung, goblet
cells in the nasopharynx, absorptive enterocytes in the gut, and proximal tubular epithelial
cells in the kidney. Even within a given cellular subtype, *ACE2* expression is heterogeneous,
with only ~1–5% of lung AT2 cells, for example, containing detectable A*CE2* mRNA. Prior
investigations have identified two distinct promoters for full-length *ACE2* which vary across
tissues in their relative usage [20], as well as a cryptic promoter driving expression of an inter-
feron-responsive truncated *ACE2* isoform [21,22]. Other studies have identified putative tran-
scription factor binding sites and epigenome signatures associated with the *ACE2* locus [23,24]
as well as transcriptome profiles associated with *ACE2* expression [25, 26].

Importantly, single cell RNA-seq approaches have intrinsic limitations for low abundance
transcripts [27], and most studies of *ACE2* mRNA have lacked validation at the protein level.
Investigations of ACE2 protein expression have been relatively limited and confounded by

uncertain specificity of different commercial ACE2 antibodies. Recently, we engineered *ACE2*-overexpressing and *ACE2*-deleted cell lines and performed systematic testing of a panel of commercial antibodies by flow cytometry, finding only 2 of 13 to exhibit specificity and sensitivity for ACE2 surface protein [28]. Unexpectedly, we found that multiple isogenic cell lines demonstrated heterogeneity of endogenous ACE2 expression, suggesting that they may serve as a simplified model to dissect the molecular pathways that govern ACE2 expression. To this end, we now report the findings of our high-throughput CRISPR screens for modifiers of endogenous ACE2 surface abundance in HuH7 and Calu-3 cells.

## Results

### CRISPR screen for HuH7 ACE2 modifiers

To identify functional regulators of ACE2 surface abundance, we first defined a list of candidate genes from several different sources (Fig 1A and S1 Table). These included modifiers of SARS-CoV-2 cytopathic effect in recently reported CRISPR screens [29–36], genes in proximity to human GWAS loci associated with COVID-19 susceptibility [37] (S2 Table), genes we previously identified by RNA-seq whose correlation was associated with *ACE2* expression in sorted HuH7 cells [28], candidates identified in our own pilot genome-wide CRISPR screens for ACE2 abundance (S1 Fig and S3 and S4 Tables), and a set of hypothesis-driven manually selected genes. In total, we targeted 833 genes with 15 gRNAs per gene. Amplicons of the pooled gRNA sequences were inserted into the pLentiCRISPRv2 construct [38] and the diversity and representation of the resulting plasmid pool were verified by deep sequencing.

We independently screened both HuH7 wild-type cells, in which ~3–5% of cells express detectable ACE2, and HuH7 cells derived by serial enrichment with 3 rounds of FACS to contain ~60–70% ACE2-positive cells (Fig 1B and 1C). Cells were then transduced at >200X coverage with the customized CRISPR library, passaged for 14 days to allow for target gene disruption and turnover of residual protein, and sorted by flow cytometry into selected populations. In the screen of wild-type cells, the ~3–5% of ACE2-positive cells were collected in one population and the median ~10% of ACE2-negative cells were collected in another. In the screen of ACE2-enriched cells, the top ~10% of cells with the greatest ACE2 abundance were collected along with the median ~10% of ACE2-negative cells. The relative abundance of every gRNA in each population was quantified by deep sequencing. Quality control analysis demonstrated improved depth of library coverage and discrimination of control gRNAs in the focused CRISPR screens relative to the primary genome-wide screens (S2 and S3 Figs). As expected, *ACE2* itself was identified among the top positive regulators of ACE2 abundance in both screens, while gRNAs targeting many other genes without a known role in ACE2 regulation exhibited significant enrichment or depletion in sorted populations (Fig 1D and 1E, S5 and S6 Tables). In total, we identified 19 high-confidence positive regulators and 16 high-confidence negative regulators of ACE2 surface abundance in HuH7 cells (FDR<0.05, absolute value of $\log_2$ fold-change >1). Analysis of gRNA representation in the FACS input samples relative to the plasmid pool revealed no bias in genes influencing cellular viability or proliferation (S4 Fig and S7 Table). Supporting the reproducibility of the identified ACE2 modifiers, there was a very high degree of concordance between the results of the independent screens of wild-type and ACE2-enriched cells (Fig 1F, $r$ = 0.84).

### Arrayed validation of HuH7 ACE2 modifiers

To validate our CRISPR screen results, we generated single gene CRISPR-targeting lentivirus constructs for 21 of the top-scoring hits from either screen and tested whether single gene disruption affected surface ACE2 levels. We found one gene, *UROD*, to be a false positive of the

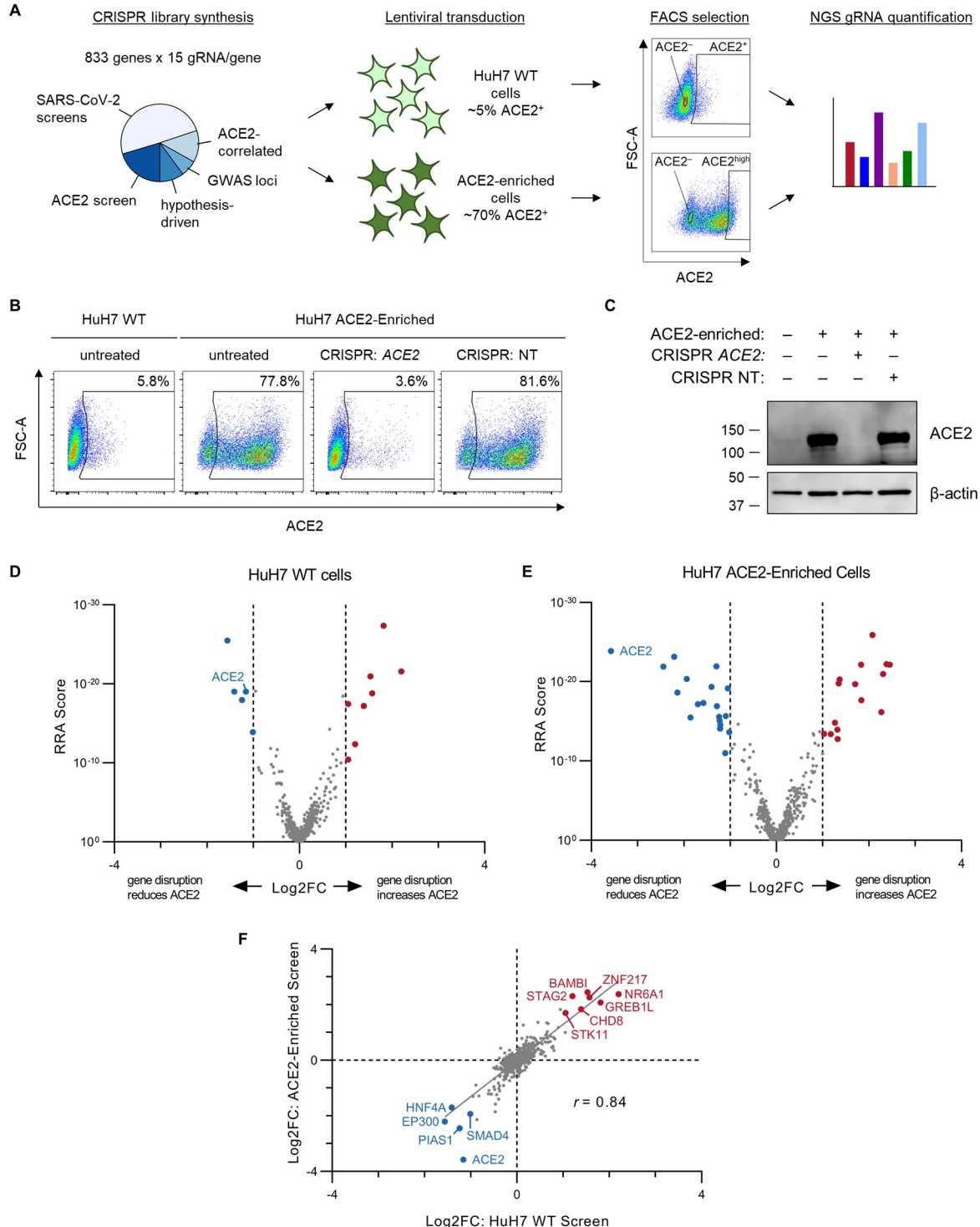

**Fig 1. CRISPR screen for ACE2 modifiers.** (A) Schematic of ACE2 CRISPR screening strategy. A total of 833 candidate ACE2 modifiers were selected from a variety of sources and targeted with a high-resolution CRISPR library containing 15 gRNAs per gene. This library was then used to screen in parallel both wild-type and ACE2-enriched HuH7 cells. Selection of individual cells was performed based on surface ACE2 abundance by FACS and individual gRNA abundance in each sorted subpopulation was quantified by massively parallel sequencing. (B) Flow cytometry plots of ACE2 surface abundance for wild-type or serially ACE2-enriched HuH7 cells, with or without transduction of a lentiviral CRISPR construct with a *ACE2*-targeting or control nontargeting (NT) gRNA. ACE2-positive gates were established on unstained wild-type cells. (C) Immunoblot for ACE2 and β-actin in lysates prepared from the same cell populations as in (B). (D-E) Volcano plots of MAGeCK Robust Rank Aggregation scores relative to gene-level gRNA log₂ fold-change for each gene tested

in the focused CRISPR library in screens of ACE2 surface abundance in HuH7 wild-type (D) or ACE2-enriched (E) cells. Genes with FDR <0.05 and absolute $\log_2$ fold-change >1 are highlighted, with positive regulators in blue and negative regulators in red. (F) Correlation of gene-level aggregate gRNA $\log_2$ fold-change between the independent secondary screens of HuH7 wild-type and ACE2-enriched cells. Genes identified in both screens with FDR <0.05 and absolute $\log_2$ fold-change >1 are highlighted and annotated.

screen due to its disruption causing increased cellular autofluorescence in the detection channel of the Alexa Fluor 647-conjugated secondary antibody. *UROD*-targeted cells exhibited increased fluorescence in this channel even when the conjugated secondary antibody was omitted (S5A Fig), and replacement of this secondary antibody with an Alexa Fluor 488-conjugated secondary antibody abrogated the difference in ACE2 signal (S5B Fig). These findings are consistent with the known role of UROD in decarboxylating uroporphyrinogen-III, as cells with accumulation of uroporphyrin have been shown to exhibit increased fluorescence between 620–680 nm [39]. Of the remaining genes, we confirmed a significant effect of single gene CRISPR-targeting on ACE2 surface staining for 17 of 20 genes in either HuH7 wild-type or ACE2-enriched cells (Fig 2 and S7 Fig, S9 Table), with none of these genes showing increased autofluorescence and each replicating the change in ACE2 abundance with an Alexa Fluor 488-conjugated secondary antibody. Consistent with the stronger enrichment and

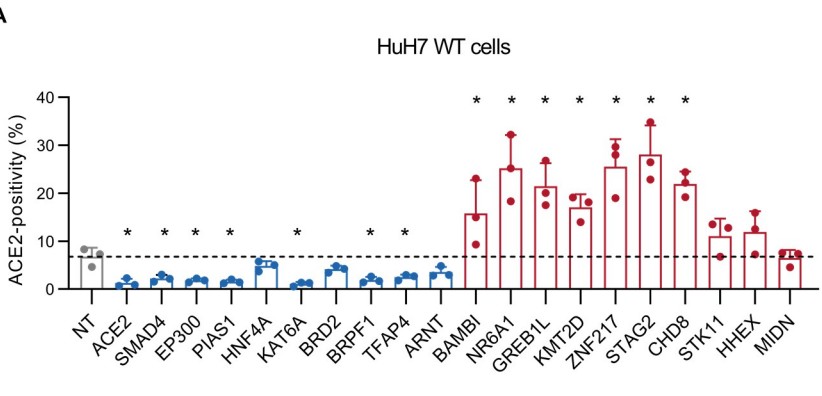

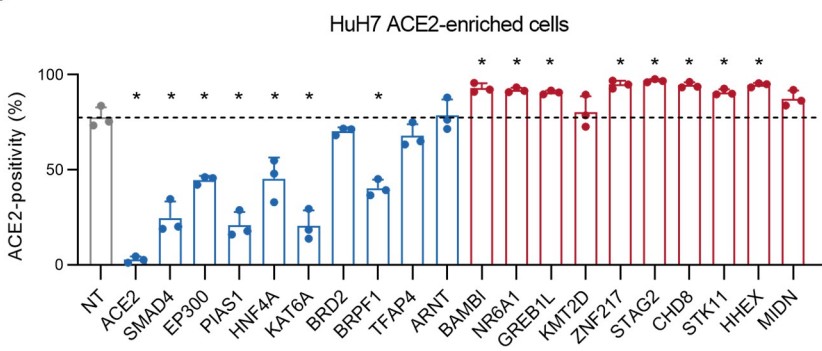

**Fig 2. Arrayed validation of ACE2 modifiers.** (A-B) Percentage of surface ACE2-positive cells after CRISPR-mediated single gene disruption of the indicated target gene or a nontargeting control on either a HuH7 parental wild-type (A) or serially ACE2-enriched background (B). Candidate genes associated with positive or negative regulation of ACE2 in the CRISPR screen are highlighted in blue and red, respectively. ACE2-positive gates were defined on control unstained HuH7 wild-type cells and the proportion of ACE2-positive cells for each population is displayed for each of 3 independent biologic replicates. Error bars depict standard deviation. Asterisks indicated p-value <0.05 relative to nontargeting controls by Student's t-test. Source data and statistical analysis are provided in S9 Table.

depletion of hits in the CRISPR screen of ACE2-enriched cells relative to wild-type cells, validation testing of these ACE2-enriched cells also exhibited increased power to detect significant changes in ACE2 surface abundance (S9 Table).

## HuH7 ACE2 modifiers are enriched for regulators of gene expression, functional networks, and viral host factors

We next analyzed the 35 ACE2 modifiers identified in our CRISPR screens for enrichment in annotated gene ontologies and protein-protein interactions. We found the greatest enrichment for several molecular functions involved in gene expression, including transcriptional regulation, chromatin binding, and DNA binding (Fig 3A). ACE2 modifiers were significantly enriched ($p < 10^{-7}$) for annotated protein-protein interactions in the STRING database (Fig 3B) [40].

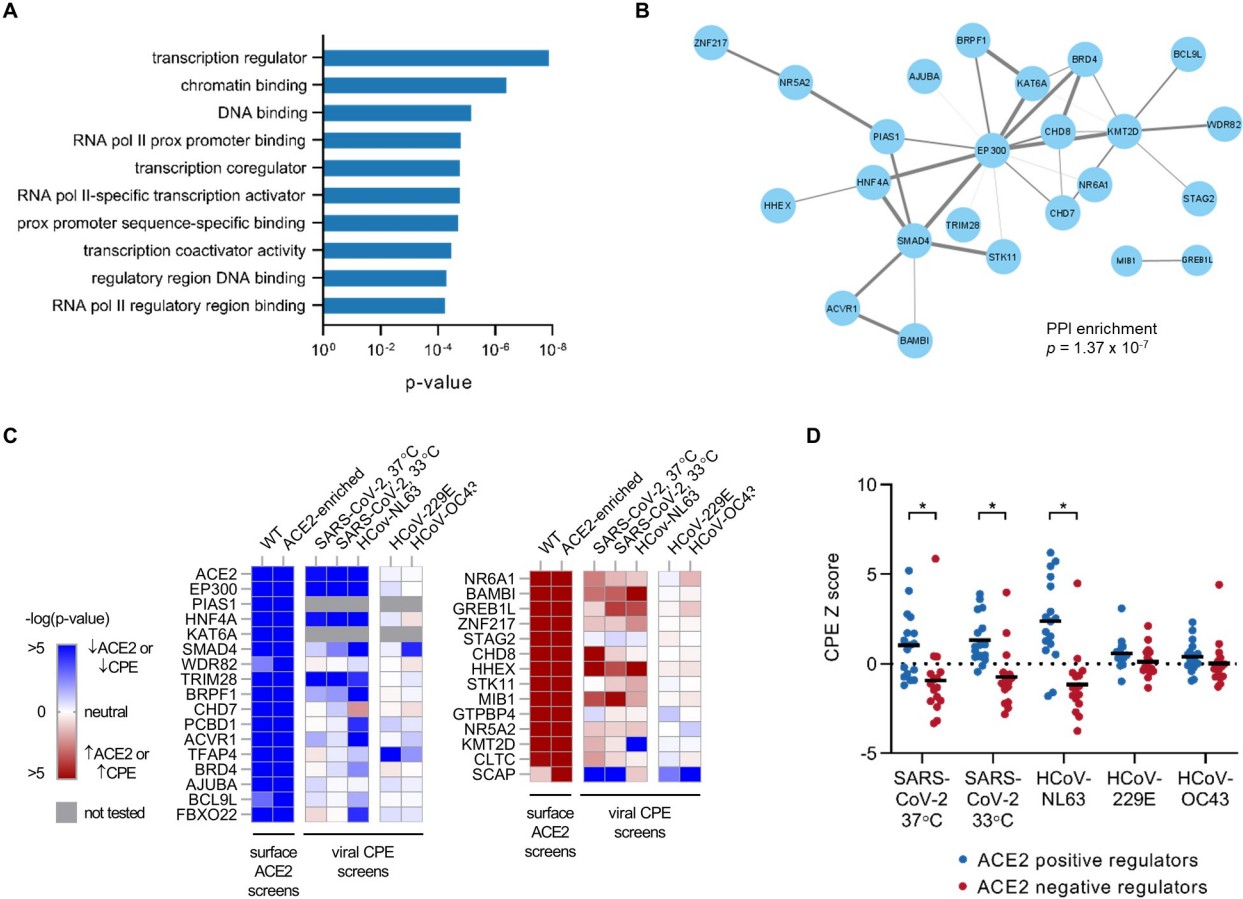

**Fig 3. Analysis of ACE2 modifiers.** (A) Top 10 molecular function ontologies enriched in genes identified as ACE2 modifiers relative to all genes tested in the focused CRISPR library. (B) Network analysis of ACE2 modifiers identified in the focused CRISPR screens of HuH7 WT or ACE2-enriched cells. Significance testing for the number of detected protein-protein interactions relative to a randomly selected gene set was calculated by STRING. Only nodes detected with a significant interaction with another node are displayed. The width of the edge between nodes is weighted by the strength of the protein-protein interaction by STRING. (C) Heat map of log-transformed p-values for each ACE2 modifier identified in this study in comparison to log-transformed p-values for cytopathic effect during infection of HuH7.5 cells with the indicated coronaviruses [35]. Viruses are grouped by those whose entry is mediated by ACE2 (SARS-CoV-2, HCoV-NL63) or by an alternative receptor (HCoV-229E, HCoV-OC43) (D) Comparison of CPE effect with the indicated coronavirus [35] for groups of genes identified in this study as ACE2 positive or negative regulators. Positive Z-scores correspond to gene disruption conferring resistance to CPE. Asterisks indicates p<0.001 by Student's t-test. A significant difference is observed for only those viruses whose cellular entry is ACE2-mediated.

We cross-referenced the data from our ACE2 CRISPR screen in HuH7 cells to that from a recently reported genome-wide CRISPR screen of HuH7.5 cytopathic effect during infection by SARS-CoV-2 or other coronaviruses [35]. As expected, *ACE2* itself was among the top genes identified both as a positive regulator of ACE2 surface abundance in HuH7 cells in our screen and as a proviral host factor for both SARS-CoV-2 and another ACE2-dependent coronavirus, HCoV-NL63, but not for other coronaviruses, HCoV-229E and HCoV-OC43, that use alternative cellular receptors (Fig 3C) [41,42]. Overall, among the genes we identified as ACE2 regulators, those that promoted ACE2 abundance in our screen were more likely to sensitize HuH7.5 cells to SARS-CoV-2 and HCoV-NL63 infection, while those that repressed ACE2 abundance were more likely to confer resistance to SARS-CoV-2 and HCoV-NL63 infection (Fig 3C and 3D). A similar correlation for ACE2 modifiers with cytopathic effect was not observed for HCoV-229E and HCoV-OC43 (Fig 3C and 3D). These findings support the relevance of our ACE2 screen to viral infection and suggest that even modest changes in ACE2 expression may influence cellular susceptibility to viral cytopathic effect.

## Cholesterol regulatory genes that influence coronavirus infection do not influence HuH7 ACE2 surface abundance

Multiple CRISPR screens have implicated cholesterol regulation as an important mediator of host cell interactions with SARS-CoV-2 [30,31,35,36]. In addition to canonical SREBP regulators, we also noted the identification of several genes in these studies that we had recently identified in a screen for regulators of low-density lipoprotein (LDL) uptake [43]. These included *RAB10* and multiple components of the exocyst complex [31,35,36]. We therefore systematically examined the overlap among these CRISPR screens for LDL uptake, viral infection, and ACE2 abundance. We observed that gene disruptions which reduced LDL uptake were more likely than those that did not to also confer cellular resistance to SARS-CoV-2 both at 37°C and at 33°C (S6A Fig). By contrast, disruption of these same genes was not associated with a significant change in ACE2 abundance in either of our screens of wild-type or ACE2-enriched HuH7 cells (S6B Fig). Further supporting an ACE2-independent effect on viral infection, positive regulators of LDL uptake did not influence cellular sensitivity to HCoV-NL63 (S6A Fig). Among these, disruption of the canonical SREBP regulators *SCAP*, *MBTPS1*, and *MBTPS2* conferred resistance to SARS-CoV-2 infection, with a similar effect for infection with the ACE2-independent HCoV-OC43 and HCoV-229E but not the ACE2-dependent HCoV2-NL63 (S6C Fig). These findings suggest that cholesterol regulatory host factors likely influence SARS-CoV-2 infection through an ACE2-independent mechanism.

## Regulation of ACE2 by *SMAD4*, *EP300*, *PIAS1*, and *BAMBI* is mediated at the mRNA level in HuH7 cells

Among the genes with the largest functional influence on ACE2 abundance in our screen and in our single gene validation experiments were *SMAD4* and multiple genes previously associated with SMAD4 signaling. These include *EP300*, encoding a histone acetyltransferase that is recruited by SMAD complexes to function as a coactivator for target genes [44, 45]; *PIAS1*, encoding an E3 sumo ligase whose substrates include SMAD4 and which has been shown to modulate SMAD4-dependent TGF-β signaling [46]; and *BAMBI*, encoding a decoy receptor that negatively regulates TGF-β signaling through SMAD4 [47]. To establish the level of regulation for ACE2 surface protein by these modifiers, we analyzed HuH7 cells individually targeted for each gene by CRISPR. To distinguish between changes in cell surface ACE2 being due either to a change in total cellular levels or to altered trafficking of ACE2, we quantified total cellular ACE2 protein abundance both by immunoblotting of cell lysates (Fig 4A) and by

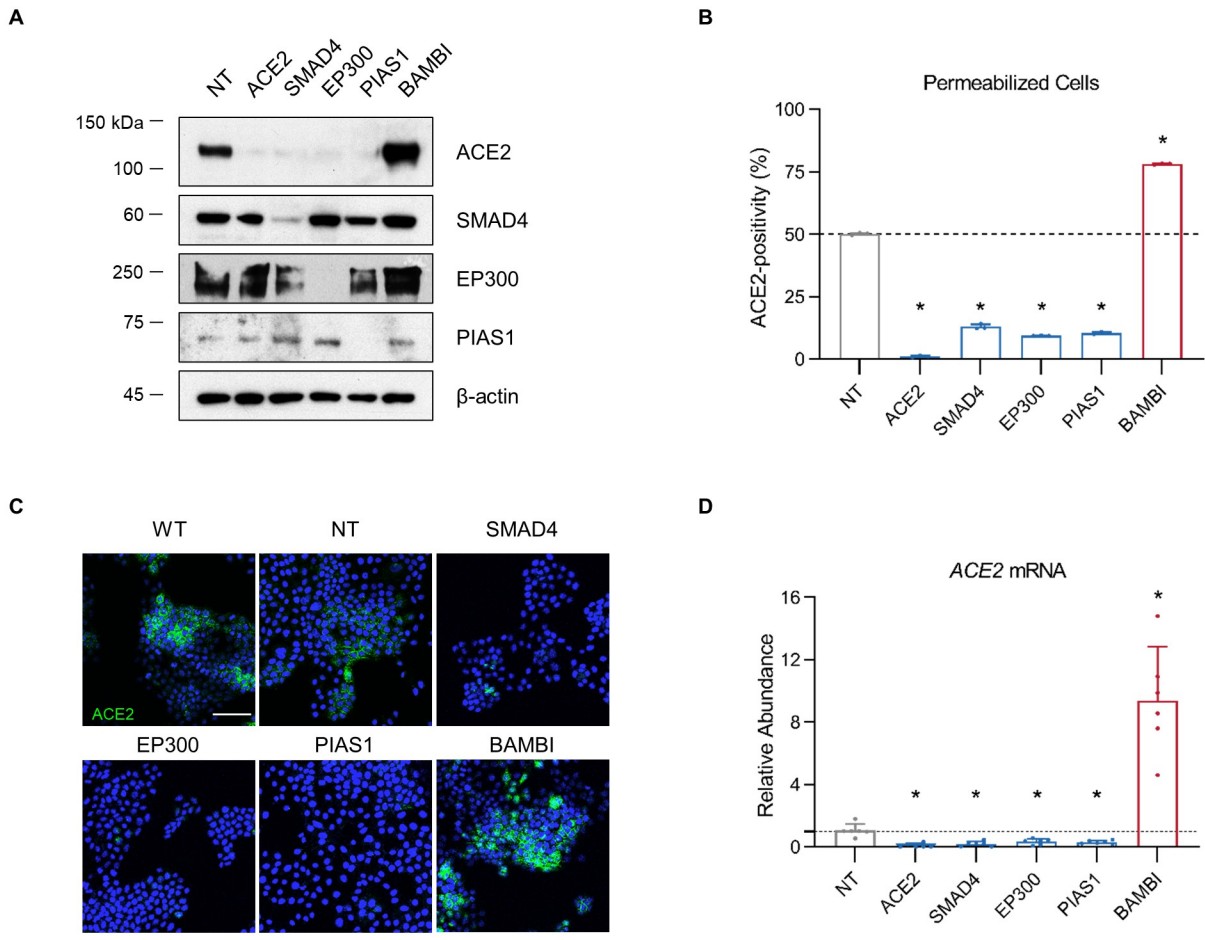

**Fig 4. Regulation of ACE2 by SMAD4, EP300, PIAS1, and BAMBI is mediated at the mRNA level.** (A) Immunoblotting of lysates collected from HuH7 cells targeted by CRISPR with a gRNA against the indicated gene or a nontargeting (NT) control sequence. (B) Proportion of saponin-permeabilized cells exhibiting ACE2 staining above background after CRISPR targeting of the indicated gene in ACE2-enriched HuH7 cells. (C) Immunofluorescence microscopy of ACE2 staining (green) in ACE2-enriched HuH7 cells either untreated (WT) or targeted by CRISPR with a gRNA against the indicated gene or a nontargeting (NT) control sequence. Scale bar = 100 μm. (D) Quantification of relative ACE2 mRNA levels in the indicated cell lines by qRT-PCR of *ACE2* mRNA, normalized to the mean Ct for a panel of control transcripts (*RPL37*, *RPL38*, and *ACTB*). Asterisks indicate p<0.05 relative to nontargeting controls by Student's t-test. Individual data points in panels B and D represent technical replicates.

flow cytometry of permeabilized cells (Fig 4B). In all cases, changes in total cellular ACE2 abundance were consistent with changes in surface ACE2 abundance. Immunofluorescence microscopy of permeabilized cells likewise demonstrated alterations in total ACE2 abundance with no striking redistribution of ACE2 in CRISPR-targeted cells (Fig 4C). Quantitative RT-PCR demonstrated changes in *ACE2* mRNA levels in CRISPR-targeted cells that were consistent with the changes in surface-displayed ACE2 protein in these cells (Fig 4D). Together, these findings indicate that ACE2 regulation by *SMAD4*, *EP300*, *PIAS1*, and *BAMBI* in HuH7 cells is mediated at the mRNA level.

## HuH7 ACE2 modifiers alter cellular susceptibility to SARS-CoV-2 infection

We next analyzed HuH7 cells with CRISPR-mediated disruption of ACE2 modifiers for their sensitivity to SARS-CoV-2 infection. We found that those CRISPR-targeted cells with

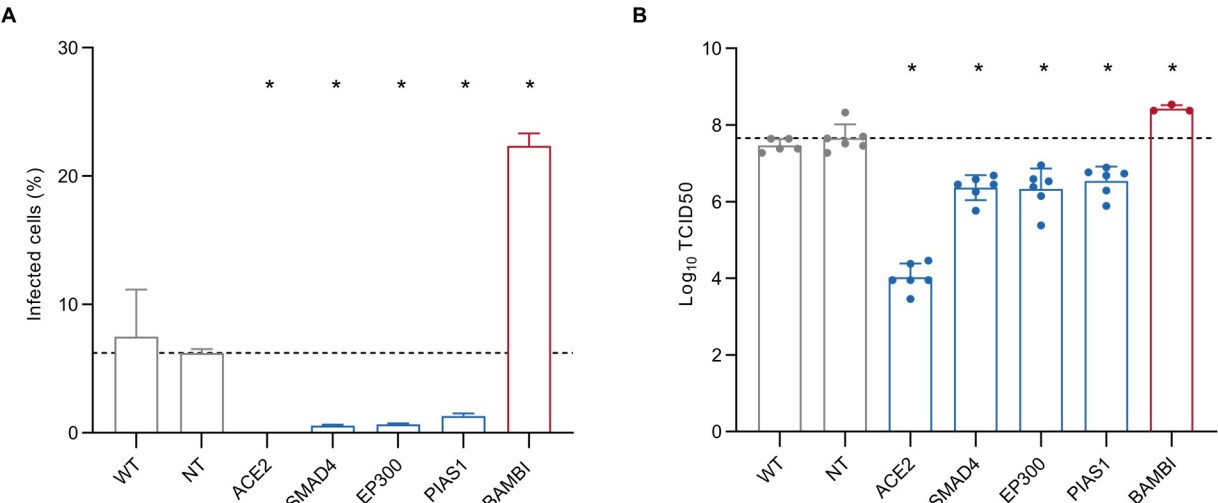

**Fig 5. ACE2 regulators influence cellular sensitivity to SARS-CoV-2 infection.** (A) Percentage of either wild-type or CRISPR-targeted cells with positive staining for SARS-CoV-2 nucleocapsid protein at 2 days post-infection with SARS-CoV-2 at MOI 1. An average of 214 fields of view (range 151–295) were analyzed for each condition over 2 independent biologic replicates. Error bars indicate standard deviation. (B) Infectious titers of cellular supernatants collected at 2 days post-infection with SARS-CoV-2 at MOI 1, plotted as $\log_{10}$(TCID50) with error bars indicating standard deviation. For both experiments, asterisks indicate $p < 0.05$ relative to nontargeting controls by Student's t-test.

decreased ACE2 expression (*SMAD4*, *EP300*, and *PIAS1*) also exhibited a decrease in both the proportion of infected cells by SARS-CoV-2 nucleocapsid protein immunofluorescence (Fig 5A) and in viral infectivity by TCID50 (Fig 5B) at 48 hours post-infection. By contrast, cells with increased ACE2 expression (*BAMBI*) exhibited an increase in the proportion of SARS-CoV-2-infected cells and viral infectious titers. These findings, together with the correlation of results from our ACE2 screen and the HuH7.5 SARS-CoV-2 screen [35], support the relevance of our identified ACE2 modifiers to SARS-CoV-2 infection.

## Functional modifiers of ACE2 surface abundance are distinct between HuH7 and Calu-3 cells

To examine the generalizability of our findings in HuH7 cells, we next tested the influence of *SMAD4* disruption on ACE2 abundance in Calu-3 cells, a respiratory epithelial cell line that has been shown to recapitulate the SARS-CoV-2 replication kinetics and type I interferon responses of primary airway epithelial cells [48]. Surprisingly, surface staining of ACE2 revealed no significant change in surface ACE2 abundance upon *SMAD4* targeting (Fig 6A). To more broadly interrogate the functional regulators of ACE2 in respiratory epithelial cells, we therefore repeated our focused CRISPR screen for ACE2 modifiers in Calu-3 cells (Fig 6B and S8 Fig, S8 Table). As was the case in HuH7 cells, we again found *ACE2* itself to be readily discriminated as the top hit of the screen. Consistent with our single gene knockout studies, we found no significant enrichment or depletion in either population for gRNAs targeting *SMAD4*, *EP300*, *PIAS1*, or *BAMBI*.

Overall, we identified only 5 genes in Calu-3 cells whose disruption was associated with a significant change in ACE2 surface abundance (FDR<0.05, absolute value of $\log_2$ fold-change >1). Of these genes, only *ACE2* itself had been found to have an effect on ACE2 surface abundance in HuH7 cells. Systematic comparison between the ACE2 screens in HuH7 and Calu-3 cells revealed no significant correlation (Fig 6C). To validate the Calu3 ACE2 screen findings, we again performed single gRNA targeting for the 5 identified genes–*ACE2*, *KDM6A*, *MOGS*,

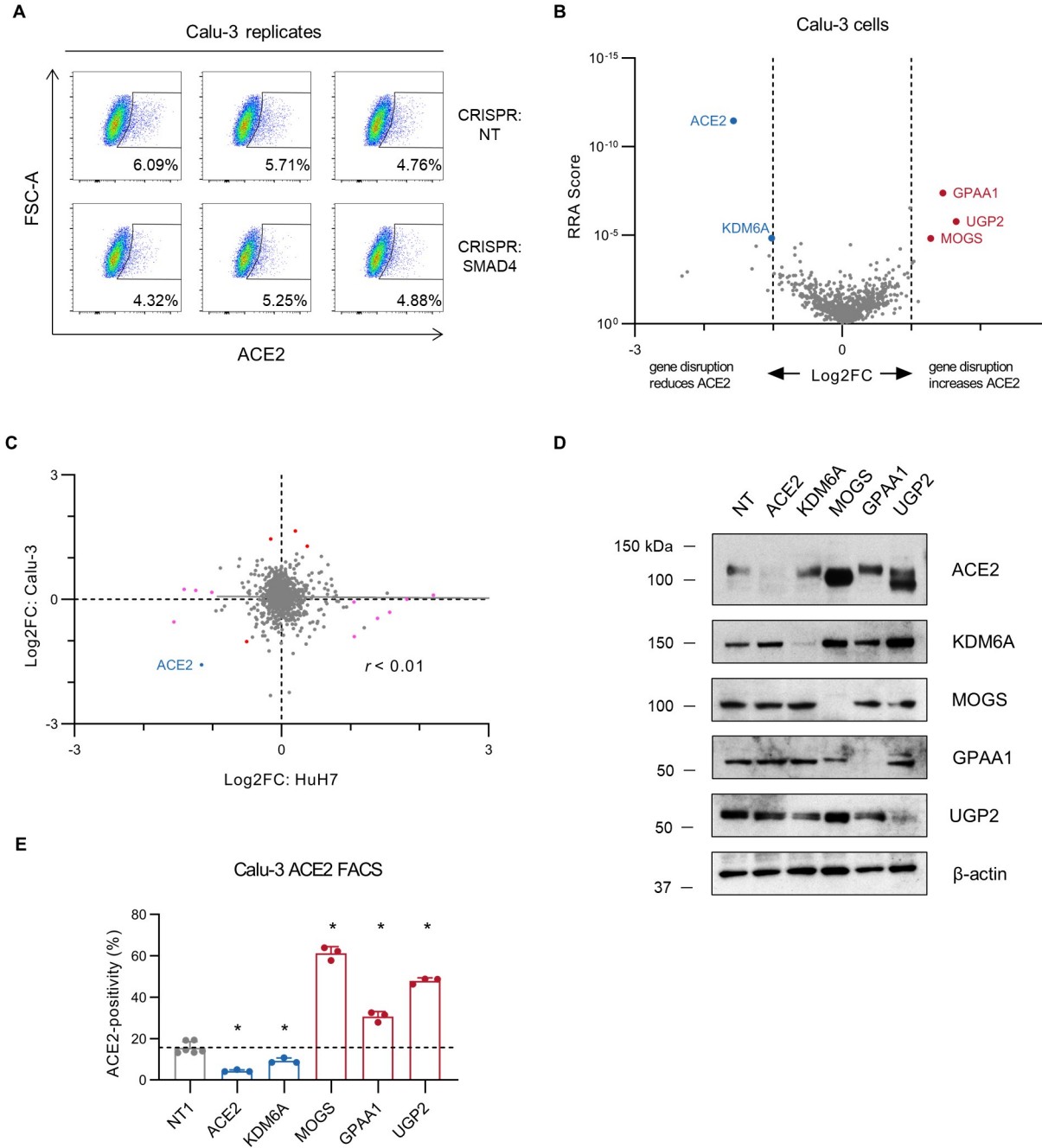

**Fig 6. Interrogation of ACE2 modifiers in Calu-3 cells.** (A) FACS plots of ACE2 surface staining in Calu-3 cells treated with control nontargeting (NT) or *SMAD4*-targeting gRNA for each of 3 technical replicates. (B) Volcano plots of MAGeCK Robust Rank Aggregation scores and gene-level gRNA log$_2$ fold-change in ACE2$^{negative}$ relative to ACE2$^{positive}$ populations for each gene tested in the focused CRISPR library in Calu-3 cells. Genes with FDR <0.05 and absolute log$_2$ fold-change >1 are highlighted, with positive regulators in blue and negative regulators in red. (C) Comparison of log$_2$ fold-change in ACE2$^{negative}$ relative to ACE2$^{positive}$ populations in focused CRISPR screens of HuH7 (x-axis) and Calu-3 (y-axis). Only *ACE2*, highlighted in blue, was identified in both screens with FDR<0.05 and absolute log$_2$ fold-change >1. Genes identified as significant in only the HuH7 or Calu-3 screens are highlighted in magenta and red, respectively. (D) Immunoblotting of lysates collected from Calu-3 cells targeted by CRISPR with a gRNA against the indicated gene or a nontargeting (NT) control sequence. (E) Percentage of cells with detectable surface ACE2 staining after CRISPR-mediated single gene disruption of the indicated target gene or a nontargeting control in Calu-3 cells. Candidate genes associated with positive or negative regulation of ACE2 in the CRISPR screen are highlighted in blue and red, respectively. ACE2-positive gates were defined on control unstained cells and the proportion of ACE2-positive cells for each population is displayed for each of 3–6 technical replicates. Error bars depict standard deviation. Asterisks indicate p<0.05 by Student's t-test. Source data and statistical analysis are provided in S9 Table.

*GPAA1*, and *UGP2*. We confirmed efficient target protein depletion by immunoblotting (Fig 6D). Flow cytometry demonstrated a significant change in ACE2 surface abundance upon each perturbation, with the direction of effect matching that observed in the CRISPR screen (Fig 6E, S9 Table). For gene disruptions with the largest effect on ACE2 surface abundance by flow cytometry (*ACE2*, *MOGS*, *UGP2*), immunoblotting of the corresponding cellular lysates for ACE2 abundance was consistent (Fig 6D). For gene disruptions with smaller effects by flow cytometry (*KDM6A* and *GPAA1*), immunoblotting was equivocal for a similar influence on total cellular ACE2 abundance, potentially due to either the reduced sensitivity of this approach for subtle changes or due to a differential effect on surface-displayed versus total cellular ACE2. Immunoblotting also revealed a change in ACE2 electrophoretic mobility in *MOGS* and *UGP2*-targeted cells, suggestive of an effect on ACE2 glycosylation status.

## Discussion

CRISPR screening is a powerful forward genetic tool for the unbiased and high-throughput functional interrogation of the human genome. This approach has been successful in the identification of host factors involved in the molecular pathogenesis for many viruses [49], including SARS-CoV-2 [29–36]. For most identified host factors, however, the molecular basis for interaction with SARS-CoV-2 remains unclear. Consistently, these screens have lent further support to the central importance of ACE2 in SARS-CoV-2 infection, with guide RNAs targeting *ACE2* among those conferring the greatest resistance to cytopathic effect. Our current study, enabled by our recently reported identification of a sensitive and specific platform for ACE2 flow cytometry [28], complements the former studies to systematically probe the genetic regulators of ACE2 surface abundance in HuH7 and Calu-3 cells.

An important caveat of our study is the focused nature of our library, limited to high-resolution functional testing of 833 candidate genes. Our initial attempts at genome-wide screening did nominate candidate ACE2 regulators that were included in our focused library, but these pilot studies were limited by an experimental bottleneck at the cell sorting stage with inadequate depth of coverage to support definitive conclusions about each gene tested. Nevertheless, for those genes interrogated by the focused library, several observations support a high degree of confidence in the functional significance of each gene to ACE2 surface abundance. First, the internal control *ACE2*-targeting gRNAs were clearly depleted in ACE2-positive cells. Second, the genes that were identified as regulating ACE2 abundance exhibited a robust statistical enrichment or depletion for several independent gRNAs. Third, we observed a very high degree of concordance in the degree of enrichment or depletion observed for each gene between the independent screens of wild-type or ACE2-enriched HuH7 cells. Fourth, identified genes were highly enriched for related functional annotations and protein-protein interactions and exhibited significant overlap with previously identified modulators of HuH7.5 cellular susceptibility to ACE2-dependent coronaviruses. Finally, of the screen hits we selected for single gene validation testing, most (17/20 in HuH7 cells, 5/5 in Calu-3 cells) were confirmed to regulate ACE2 surface abundance.

Given the sensitivity of our screens to detect subtle influences on ACE2 abundance, a number of negative findings are noteworthy. Cholesterol regulatory genes in the SREBP pathway have been implicated in SARS-CoV-2 infection, but we did not detect an association with ACE2 abundance either by focused analysis of canonical SREBP regulators or by systematic comparison to all HuH7 LDL uptake regulators identified in our previously reported genome-wide CRISPR screen [43]. These findings are consistent with the association of SREBP regulators with ACE2-independent coronaviruses and the lack of association with ACE2-dependent HCoV-NL63. Angiotensin receptor blockade has been postulated to increase surface ACE2

expression, raising concern for potential increased COVID-19 risk for patients on ACE inhibitors or angiotensin receptor blockers. We did not however detect any association of *AGTR1* or *AGTR2* with ACE2 abundance. We also did not observe an effect for disruption of the Ang [1–7] receptor *MAS1*, arguing against feedback regulation of this axis in these cells. Although estrogen regulation of ACE2 expression has been proposed to mediate the sexual dimorphism of COVID-19 susceptibility, and estrogen receptor binding sites have been identified near the ACE2 locus [26], we did not detect an effect of *ESR1* or *ESR2* disruption on ACE2 abundance. We also did not detect a large effect for any candidate genes in proximity to variants associated with COVID-19 susceptibility by GWAS. Each of these negative results should be interpreted with caution, however, both because of the potential cell type specificity of ACE2 regulation and the intrinsic limitation of CRISPR screens for functionally redundant genes, essential genes, or compensatory mechanisms in mutant cells.

SMAD4, which we identified as a positive regulator of ACE2 gene expression in HuH7 cells, is a common mediator of TGF-β signaling [50]. In the canonical TGF-β signaling pathway, ligand binding signals through receptor-regulated SMADs that complex with SMAD4 to trigger its translocation into the nucleus and binding to specific DNA regulatory elements. Identification of the upstream signals driving SMAD4-dependent ACE2 expression in HuH7 cells is complicated by the diversity of TGF-β superfamily receptors and ligands including bone morphogenic proteins, nodal, and activin. Our screen did identify *ACVR1*, encoding activin A receptor type 1, as a positive regulator of ACE2, and this gene was similarly identified as proviral for SARS-CoV-2 and HCoV-NL63 in HuH7.5 cells [35]. The limited scope of our screen, however, precludes a comprehensive cataloguing of other potential upstream mediators of SMAD4-dependent ACE2 expression. Supporting the physiologic relevance of these findings, a recent study identified ChIP-seq peaks for SMAD4 in intestinal epithelial cells that overlapped putative enhancer regions near the murine *Ace2* locus, and furthermore detected reduced *Ace2* intestinal transcript levels upon tissue-specific *Smad4* deletion [51].

Our findings are consistent with a growing body of evidence supporting the cell type specificity of genetic interactions relevant to SARS-CoV-2 infection. Although *ACE2* itself was the top hit in each of our screens of HuH7 or Calu-3 cells, there was otherwise very little concordance between these screens. Independent CRISPR screens of SARS-CoV-2 cytopathic effect in different cell types have likewise identified limited overlap between studies. Although we observed significant overlap between ACE2 modifiers we identified in HuH7 cells with recently identified SARS-CoV-2 infection modifiers in HuH7.5 cells [35], we observed minimal overlap with SARS-CoV-2 infection modifiers identified in similar genome-scale screens of other cell lines [29,30,32,34,36]. For some of these other studies, this discrepancy may be attributable to their use of cell lines with heterologous overexpression of ACE2 [30,32,36]. In other cases, however, SARS-CoV-2 infection was tested in cell lines with endogenous ACE2 expression [29, 34] and yet still little overlap was observed. For example, we did not detect an effect of *HMGB1* disruption on ACE2 abundance in either HuH7 or Calu-3 cells, despite this gene being previously implicated in SARS-CoV-2 cytopathic effect in Vero E6, HuH7.5, and Calu-3 cells and in *ACE2* mRNA and protein expression in Vero E6 cells. It is possible that this discrepancy may be related to *HGMB1* exerting pleiotropic ACE2-dependent and ACE2-independent effects on SARS-CoV-2 infection in different cell lines. It is also possible that this and other discrepancies across screens may be related to technical differences in culture conditions, CRISPR library design, infection parameters, or functional readout. For example, another SARS-CoV-2 screen of HuH7 cells [34] did not identify *ACE2*, which was attributed to low levels of ACE2 expression and implies that the conditions of this screen would not enable the detection of ACE2 positive regulators. These technical considerations aside, our findings

strongly suggest that ACE2 regulatory networks are highly cell type specific, a conclusion that is further supported by small molecule inhibitor studies [52] and by the tissue-specific patterns of *ACE2* promoter usage [20]. The molecular basis for this cell type specificity is not established by our study but may be related to the differential activity of signaling pathways and/or *ACE2 cis*-regulatory elements in distinct tissues.

In summary, we have applied a functional genomic approach to dissect the regulatory networks of ACE2 protein expression. We have identified many previously unrecognized genetic modifiers of ACE2 expression and a putative mechanism for genes previously implicated in SARS-CoV-2 infection. Our findings highlight the critical influence of cell type on ACE2 regulation and nominate pathways for host-targeted therapeutic development.

## Materials and methods

### ACE2 CRISPR screens

Wild-type HuH7 and Calu-3 cells and serially ACE2-enriched HuH7 cells were independently screened. For each independent 3–4 biologic replicates for each cell line, a total of ~100 million cells were transduced with the GeCKOv2 library [38] or ~25 million cells were transduced with the focused CRISPR library at an MOI of ~0.3. Puromycin was added at a concentration of 3 µg/mL at day 1 post-transduction and maintained until selection of control uninfected cells was complete. Cells were passaged as needed to maintain logarithmic phase growth with total cell number maintained at all stages above a minimum of ~25 million cells for the primary screens or ~8 million cells for the focused screens. At 14 days post-transduction, edited cells (~200 million for the primary screens, ~50 million for the focused screens) were harvested and stained for surface ACE2 abundance as previously described with ACE2 antibody (R&D Systems #MAB9332) at 1:50 dilution in FACS buffer (PBS supplemented with 2% FBS) and Alexa Fluor 647-conjugated goat anti-rabbit IgG secondary antibody (AlexaFluor647 goat anti-rabbit IgG (Fisher #A32733) at 1:500 dilution in FACS buffer. Flow cytometry gates were defined for cell viability by exclusion of SYTOX Blue Dead Cell Stain (Fisher, S34857) and for ACE2 expression by comparison to unstained or ACE2-targeted control cells. For the screen of wild-type HuH7 or Calu-3 cells, the ~3–10% of the total cells that were positive for ACE2 staining (as defined by gating of unstained cells) were collected. For the screen of ACE2-enriched HuH7 cells, the brightest 10% ACE2-positive cells were collected. For each screen, a gate of the median 10% of ACE2-negative cells was also collected. A total of ~20 million cells of FACS input were also collected for comparison to the starting plasmid pool. Genomic DNA was extracted and gRNA sequences amplified and sequenced as previously described, with pooling of samples performed after barcoding of PCR amplicons [43].

### CRISPR screen analysis

FASTQ files were processed by PoolQ (Broad Institute; https://portals.broadinstitute.org/gpp/public/software/poolq) to map individual sequencing reads to reference gRNA sequences with deconvolution by barcode. Cumulative distribution functions of gRNA representation were generated by plotting normalized read counts of each gRNA against its relative rank for a given barcode. Library coverage was assessed by the proportion of gRNAs in the library detected in the sequencing run and library skewing was assessed by a ratio of the 90th and 10th percentile most abundant reads. Individual gRNA-level and aggregate gene-level enrichment analysis was performed using MAGeCK [53]. Q-Q plots were generated by plotting log-transformed observed p-values (calculated by MAGeCK gene-level analysis) against expected p-values (determined by the relative rank of each gene among the library). Genes were considered screen hits if they were identified with a MAGeCK-calculated false discovery rate < 0.05 and

absolute $\log_2$ fold-change $> 1$ in either analysis of HuH7 wild-type or ACE2-enriched cells. Enrichment of molecular functions among screen hits relative to all genes in the focused library was performed using GOrilla [54]. Gene network analysis of secondary screen hits was performed using the STRING database [40] with default settings for query of the full STRING network with medium confidence scoring threshold (0.400) and FDR stringency (0.05). Visualization of STRING interactions was performed with Cytoscape [55]. Heatmaps were generated with GraphPad Prism v.9.1.0 using log-transformed p-values for effects on HuH7 ACE2 surface abundance in this study (S5 Table), HuH7.5 viral cytopathic effect in S1 Table of Schneider et al. [35], and HuH7 LDL endocytosis in S3 Table our prior study [43].

## Design and synthesis of focused CRISPR library

Candidate genes were selected from (i) top candidate genes identified in our primary genome-wide CRISPR screen for ACE2 surface abundance; (ii) genes identified in CRISPR screens as candidate modifiers of SARS-CoV-2 cytopathic effect [29–36]; (iii) genes whose expression was correlated with ACE2 surface abundance in HuH7 cells [28]; (iv) genes in proximity to loci associated with COVID-19 susceptibility by GWAS (described below); (v) candidate genes identified in our own pilot CRISPR screens of ACE2 surface abundance in ACE2-overexpressing HEK293T cells or Caco2 cells or SARS-CoV-2 cytopathic effect in Caco2 cells; and (vi) hypothesis-driven manually selected genes. For each gene, a total of 15 optimized gRNA sequences were identified using the Broad Genetic Perturbation Platform [56]. For identification of COVID-19 GWAS loci, data freeze 4 of the COVID-19 Host Genetics Initiative was used [37]. Lead SNPs were selected from the analysis of hospitalized COVID-19 patients relative to population controls. For each indicated SNP, genes were selected based on their physical proximity to the SNP and by their Polygenic Priority Score [57] (S2 Table). Flanking sequences were appended to facilitate PCR amplification and oligonucleotides were synthesized by CustomArray (Bothell, WA). DNA assembly of the secondary library plasmid pool was performed with 125 ng of PCR amplicon and 825 ng of BsmBI-digested pLentiCRISPRv2 in a total reaction volume of 100 μL with HiFI DNA Assembly Mix (NEB) for 30 min at 50˚C. Assembly products were purified with a QIAquick PCR purification kit (Qiagen, Hilden, Germany) and 5 electroporations were performed into Endura electrocompetent cells (Lucigen, Middleton WI) and plated onto 24.5 cm$^2$ LB-agar plates. After 14 hr at 37˚C, bacteria were harvested and plasmid DNA purified with an EndoFree Plasmid Maxi kit (Qiagen). Dilution plates of electroporated cells confirmed a colony count of >100X relative to the size of the gRNA library and representation was confirmed by sequencing on an Illumina MiSeq with gRNA mapping and cumulative distribution function analysis as described above. Lentiviral stocks were generated by cotransfection of the lentiviral plasmid pool with psPAX2 and pVSVG into HEK293T cells, harvesting of supernatants, and titering of virus stocks as previously described [58].

## Arrayed validation of ACE2 modifiers

For each gene tested, a single gRNA was selected from the 15 in the library based on its degree of enrichment or depletion in the secondary screens. Each individual gRNA (S10 Table) was ligated into *BsmBI*-digested pLentiCRISPRv2 [38] and lentiviral stocks generated and titered as previously described [58]. HuH7 wild-type and ACE2-enriched cells were transduced in parallel with each lentiviral construct, treated with puromycin 3 μg/mL until no surviving cells remained among control non-transduced cells, and passaged to maintain logarithmic phase growth. Surface ACE2 abundance in nonpermeabilized cells was quantified at day 14 post-transduction by flow cytometry as previously described [28].

## Analysis of ACE2 mRNA and protein

Immunoblotting of RIPA lysates was performed as previously described [28] with antibodies against ACE2 (GeneTex, Irving CA, #GTX01160, 1:1000), β-actin (Santa Cruz Biotechnology, Dallas TX, #sc-47778, 1:5000), SMAD4 (Abcam, Cambridge UK, ab40759, 1:1000), EP300 (Abcam, ab275378, 1:500), PIAS1 (Cell Signaling Technology, Danvers MA, 3550S, 1:1000), KDM6A (Cell Signaling Technology, 33510S, 1:500), MOGS (Proteintech, Rosemont IL, 17859-1-AP, 1:500), GPAA1 (GeneTex, Irvine CA, GTX115131, 1:500), and UGP2 (Abcam, ab154817, 1:500). For Fig 4B, permeabilization prior to ACE2 flow cytometry was performed by incubating cells prior to blocking in 0.1% saponin in FACS buffer for 10 min. For all other flow cytometry experiments, nonpermeabilized cells were tested. Immunoflourescence microscopy of ACE2 was performed on ACE2-enriched HuH7 cells seeded in 35 mm poly-D lysine-coated glass bottom dishes (MatTek, Ashland MA, P35GC-1.5-14-C) as previously described [28], with random fields of view selected by an operator who was blinded to sample identity. Quantification of ACE2 transcript levels was performed by preparing total RNA from $2–4×10^6$ cells for each sample using the RNeasy Plus Micro kit (Qiagen, Hilden Germany, #74034). For qRT-pCR, cDNA was prepared using the SuperScript III first-strand synthesis kit (Thermo Fisher, #18080051), amplified with Power SYBR Green PCR Master Mix (Thermo Fisher, # 4367659), and analyzed by QuantStudio 5 Real-Time PCR (Thermo Fisher). Relative *ACE2* transcript levels (primers: 5′-AAACATACTGTGACCCCGCAT-3′ and 5′-CCAAGCCT CAGCATATTGAACA-3′) were normalized against the mean Ct value for a panel of loading controls–*RPL37* (primers: 5′-CGCAGATTCAGGCATGGAT-3′ and 5′-AGCTGCCCTCTTG GGTTTAGG-3′), *RPL38* (primers: 5′-TGCTGCTTGCTGTGAGTGTCT-3′ and 5′-CGCGG ACCAGGACCTTT-3′), and *ACTB* (5′-TTCCTTCCTGGGCATGGA-3′ and 5′-CGTCACAC TTCATGATGGAGTTG -3′). A total of 3–6 technical replicates were performed for each sample.

## SARS-CoV-2 infection assays

SARS-CoV-2 WA1 strain was obtained by BEI resources and was propagated in Vero E6 cells. Lack of genetic drift of our viral stock was confirmed by deep sequencing. Viral titers were determined by TCID50 assays in Vero E6 cells (Reed and Muench method) by microscopic scoring. All experiments using SARS-CoV-2 were performed at the University of Michigan under Biosafety Level 3 (BSL3) protocols in compliance with containment procedures in laboratories approved for use by the University of Michigan Institutional Biosafety Committee (IBC) and Environment, Health and Safety (EHS). For the immunofluorescence-mediated assay, 384-well plates (Perkin Elmer, 6057300) were seeded with HuH7 cells at 3000 cells per well and allowed to adhere overnight. Plates were then transferred to BSL3 containment and infected with SARS-CoV-2 WA1 at a multiplicity of infection (MOI) of 1. Two days post-infection, cells were fixed with 4% PFA for 30 minutes at room temperature, permeabilized with 0.3% Triton X-100, and blocked with antibody buffer (1.5% BSA, 1% goat serum and 0.0025% Tween 20). The plates were then sealed, surface decontaminated, and transferred to BSL2 for staining with antibody against SARS-CoV-2 nucleocapsid protein (Antibodies Online, Cat# ABIN6952432) overnight at 4°C followed by staining with Alexa Fluor 647-conjugated secondary antibody (goat anti-mouse, Thermo Fisher, A21235) and DAPI (Thermo Fisher). Plates were imaged with the Thermo Fisher CellInsight CX5 High-Content Screening (HCS) Platform with a 10X/0.45NA LUCPlan FLN objective and analyzed with a Cell Profiler pipeline. An average of ten fields per well were acquired and each condition was run with ten technical replicates (10 wells) over two biological experiments. Percentage of infected cells was calculated based on SARS-CoV-2 nucleocapsid

protein and DAPI staining, as previously described [59]. For the infectivity assay, HuH7 were seeded at 3 x $10^5$ cells per well in a 12-well plate and allowed to adhere overnight. The next day, cells were infected with SARS-CoV-2 WA1 at a MOI of 1 at 37˚C for 1hr. Viral inoculum was removed by 3 rounds of washing. Two days after infection, cells were harvested by scraping the monolayers and lysates were centrifuged at high speed for 5 minutes to allow the release of intracellular viral progeny in infected cells. Infectious titer of the supernatants was determined by TCID50 assay.

## Supporting information

**S1 Fig. Primary genome-wide CRISPR screen for ACE2 modifiers.** Manhattan plots demonstrating gene-level MAGeCK Robust Rank Aggregation (RRA) scores for each gene in the GeCKOv2 library plotted according to its chromosomal transcription start site. N = nontargeting controls. Plots are displayed for both positive regulation (A, C; gRNAs depleted in ACE2-positive relative to ACE2-negative populations) and negative regulation (B, D; gRNAs enriched in ACE2-positive relative to ACE2-negative populations) for independent screens of HuH7 wild-type cells (A, C) and HuH7 ACE2-enriched cells (B, D). The top 10 genes for each analysis are highlighted in red.
(TIF)

**S2 Fig. Quality analysis of genome-wide HuH7 ACE2 CRISPR screens.** (A-B) Cumulative distribution functions of normalized read counts for each gRNA in ACE2-sorted populations in 4 independent biologic replicates of the genome-wide primary CRISPR screen of wild-type (A) and ACE2-enriched (B) HuH7 cells. (C) Mean $\log_2$ fold-change of each individual gRNA targeting ACE2, the aggregate mean of these individual ACE2-targeting gRNAs, and a violin plot of the distribution for all 1000 control nontargeting gRNAS (lines indicate quartiles) in the genome-wide ACE2 CRISPR screens of wild-type (C) and ACE2-enriched (D) HuH7 cells. (E-H) Q-Q plots of observed versus expected–log(p) for positive (E,G) or negative (F, H) regulation of ACE2 surface abundance for every gene in the primary genome-wide CRISPR screen of HuH7 wild-type (E-F) or ACE2-enriched (G-H) cells. Observed p-values were calculated by MAGeCK gene-level analysis.
(TIF)

**S3 Fig. Quality analysis of focused HuH7 ACE2 CRISPR screens.** (A-B) Cumulative distribution functions of normalized read counts for each gRNA in ACE2-sorted populations in 3 independent biologic replicates of the focused secondary CRISPR screen of wild-type (A) and ACE2-enriched (B) HuH7 cells. In each subpopulation, >99.9% of all possible gRNAs were detected and the ratio between the 90th and 10th percentile most abundant gRNAs was <10. (C-D) Mean $\log_2$ fold-change of each individual gRNA targeting ACE2, the aggregate mean of these individual ACE2-targeting gRNAs, and a violin plot of the distribution for all 75 control nontargeting gRNAS (lines indicate quartiles) in the focused ACE2 CRISPR screens of wild-type (C) and ACE2-enriched (D) HuH7 cells. (E-H) Q-Q plots of observed versus expected–log(p) for positive (E,G) or negative (F, H) regulation of ACE2 surface abundance for every gene in the focused CRISPR screen of HuH7 wild-type (E-F) or ACE2-enriched (G-H) cells. Observed p-values were calculated by MAGeCK gene-level analysis.
(TIF)

**S4 Fig. Enrichment analysis of gRNA representation prior to FACS selection.** Individual gRNAs were quantified in the starting CRISPR library plasmid pool and the HuH7 wild-type screen day 14 FACS input. A violin plot is displayed for all genes in the library, along with individual data points for genes identified as Tier 1 Core Essential Genes in human cell lines [60]

or identified as HuH7 ACE2 modifiers in this study. Lines indicate median values. Annotated p-values were calculated by Fisher's LSD test.
(TIF)

**S5 Fig. Disruption of *UROD* increases cellular autofluorescence.** (A-B) FACS plots of fluorescence intensities of control and *UROD*-targeted HuH7 cells either unstained or incubated with an ACE2 antibody and a corresponding Alexa Fluor 488 (A) or Alexa Fluor 647 (B) conjugated secondary antibody.
(TIF)

**S6 Fig. Comparison of CRISPR screen results for modifiers of HuH7 LDL endocytosis, HuH7.5 SARS-CoV-2 infection, and HuH7 ACE2 surface abundance.** (A) The subset of genes tested by high-resolution CRISPR screening for both LDL endocytosis and ACE2 surface abundance were divided into those whose disruption reduced LDL uptake (n = 15) and "other genes" whose disruption did not reduce LDL uptake (n = 55). The Z-score for each gene within each group was then compared for each of the viral CPE screens reported by Schneider et al. Asterisks indicate $p < 0.05$ by Student's t-test. (B) Genes were grouped as in (A) and compared for their association with surface ACE2 abundance in secondary screens of either HuH7 wild-type or ACE2-enriched cells. (C) Heat maps for log-transformed p-values of canonical SREBP regulators *LDLR*, *SCAP*, *MBTPS1*, and *MBTPS2* in screens of HuH7 LDL uptake [43] and ACE2 surface abundance (this study) and HuH7.5 SARS-CoV-2 cytopathic effect [35].
(TIF)

**S7 Fig. Arrayed validation of HuH7 ACE2 modifiers.** FACS plots of gating strategy and ACE2 staining relative to forward scatter area for untreated and single gRNA CRISPR-targeted wild-type HuH7 cells. FACS plots for a representative replicate of Fig 2A are displayed. Raw data for all replicates and statistical analysis are provided in S9 Table.
(TIF)

**S8 Fig. Quality analysis of Calu-3 ACE2 CRISPR screen.** (A) Cumulative distribution functions of normalized read counts for each gRNA in ACE2-sorted populations in 3 independent biologic replicates of the focused CRISPR screen of Calu-3 cells. (B) Mean $\log_2$ fold-change of each individual gRNA targeting *ACE2*, the aggregate mean of these individual ACE2-targeting gRNAs, and a violin plot of the distribution for all 75 control nontargeting gRNAS (lines indicate quartiles). (C-D) Q-Q plots of observed versus expected–log(p) for positive (C) or negative (D) regulation of ACE2 surface abundance for every gene in the focused CRISPR screen of Calu-3 cells. Observed p-values were calculated by MAGeCK gene-level analysis.
(TIF)

**S9 Fig. Arrayed validation of Calu-3 ACE2 modifiers.** FACS plots of gating strategy and ACE2 staining relative to forward scatter area for untreated and single gRNA CRISPR-targeted Calu-3 cells. FACS plots for a representative replicate of Fig 6E are displayed. Raw data for all replicates and statistical analysis are provided in S9 Table.
(TIF)

**S1 Table. CRISPR library design.** Primary sources and selection criteria are indicated for the gene lists that comprise the focused CRISPR library of potential ACE2 modifiers.
(XLSX)

**S2 Table. GWAS candidate gene identification.** Top-scoring SNPs associated with COVID-19 infection are listed along with candidate causal genes selected by either their physical

proximity or Polygenic Prioritization Score [57].
(XLSX)

**S3 Table. Genome-wide ACE2 CRISPR screen of HuH7 wild-type cells.** MAGeCK output for gene-level and individual gRNA-level enrichment or depletion in ACE2-positive cells relative to ACE2-negative cells. Negative $\log_2$-fold change and RRA scores indicate gRNA depletion in ACE2-positive cells (gene disruption associated with reduced ACE2 abundance).
(XLSB)

**S4 Table. Genome-wide ACE2 CRISPR screen of HuH7 ACE2-enriched cells.** MAGeCK output for gene-level and individual gRNA-level enrichment or depletion in ACE2-high cells relative to ACE2-negative cells. Negative $\log_2$-fold change and RRA scores indicate gRNA depletion in ACE2-high cells (gene disruption associated with reduced ACE2 abundance).
(XLSB)

**S5 Table. Focused CRISPR screen of HuH7 wild-type cells.** MAGeCK output for gene-level and individual gRNA-level enrichment or depletion in ACE2-positive cells relative to ACE2-negative cells. Negative $\log_2$-fold change and RRA scores indicate gRNA depletion in ACE2-positive cells (gene disruption associated with reduced ACE2 abundance).
(XLSX)

**S6 Table. Focused ACE2 CRISPR screen of HuH7 ACE2-enriched cells.** MAGeCK output for gene-level and individual gRNA-level enrichment or depletion in ACE2-high cells relative to ACE2-negative cells. Negative $\log_2$-fold change and RRA scores indicate gRNA depletion in ACE2-high cells (gene disruption associated with reduced ACE2 abundance).
(XLSX)

**S7 Table. Analysis of gRNA enrichment in FACS input relative to plasmid pool for HuH7 wild-type focused CRISPR screen.** MAGeCK output for gene-level gRNA enrichment or depletion in the HuH7 wild-type ACE2 screen, comparing starting plasmid pool to FACS input on day 14 of the screen. Negative $\log_2$-fold change and RRA scores indicate gRNA depletion in day 14 samples (gene disruption reduces cellular fitness).
(XLSX)

**S8 Table. Focused ACE2 CRISPR screen of Calu-3 cells.** MAGeCK output for gene-level and individual gRNA-level enrichment or depletion in ACE2-positive cells relative to ACE2-negative cells. Negative $\log_2$-fold change and RRA scores indicate gRNA depletion in ACE2-high cells (gene disruption associated with reduced ACE2 abundance).
(XLSX)

**S9 Table. Arrayed validation of ACE2 modifiers.** Individual replicate values of ACE2-positivity and normalized ACE2 mean fluorescence intensity in HuH7 WT or ACE2-enriched cells or Calu-3 cells transduced with a lentiCRISPR construct targeting the indicated gene or a control nontargeting (NT) sequence. Statistical testing was performed by Student's t-test of 2-tailed distributions assuming equal variance.
(XLSX)

**S10 Table. Individual gRNA sequences used for single gene lentiCRISPR targeting.** The indicated gRNA sequences were selected based on their enrichment or depletion in CRISPR screens for ACE2 abundance.
(XLSX)

## Author Contributions

**Conceptualization:** Emily J. Sherman, Brian T. Emmer.

**Data curation:** Emily J. Sherman, Carmen Mirabelli, Vi T. Tang, Taslima G. Khan, Kyle Leix, Andrew A. Kennedy, Sarah E. Graham, Brian T. Emmer.

**Formal analysis:** Emily J. Sherman, Carmen Mirabelli, Vi T. Tang, Taslima G. Khan, Kyle Leix, Andrew A. Kennedy, Sarah E. Graham, Cristen J. Willer, Andrew W. Tai, Jonathan Z. Sexton, Christiane E. Wobus, Brian T. Emmer.

**Funding acquisition:** Brian T. Emmer.

**Writing – original draft:** Emily J. Sherman, Brian T. Emmer.

**Writing – review & editing:** Emily J. Sherman, Carmen Mirabelli, Vi T. Tang, Taslima G. Khan, Kyle Leix, Andrew A. Kennedy, Sarah E. Graham, Cristen J. Willer, Andrew W. Tai, Jonathan Z. Sexton, Christiane E. Wobus, Brian T. Emmer.

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
