## [Decision Letter · Decision Letter 0]

24 Oct 2021

Dear Dr. Emmer,

Thank you very much for submitting your manuscript "Identification of ACE2 modifiers by CRISPR screening" for consideration at PLOS Pathogens. As with all papers reviewed by the journal, your manuscript was reviewed by members of the editorial board and by several independent reviewers. In light of the reviews (below this email), we would like to invite the resubmission of a significantly-revised version that takes into account the reviewers' comments.

The reviewers had a number of constructive comments that will make the manuscript stronger once addressed. Importantly, it will be necessary to perform some validation on another cell line/type with a subset of the important hits identified in HUH-7 cells in a revised manuscript, in addition to addressing many of the comments the reviewers had about the published literature and the data in the manuscript.

We cannot make any decision about publication until we have seen the revised manuscript and your response to the reviewers' comments. Your revised manuscript is also likely to be sent to reviewers for further evaluation.

Sincerely,

Stacy M Horner

Associate Editor

PLOS Pathogens

Andrew Pekosz

Section Editor

PLOS Pathogens

Kasturi Haldar

Editor-in-Chief

PLOS Pathogens

orcid.org/0000-0001-5065-158X

Michael Malim

Editor-in-Chief

PLOS Pathogens

orcid.org/0000-0002-7699-2064

Reviewer's Responses to Questions

**Part I - Summary**

Reviewer #1: In this manuscript, Sherman and co-workers present a CRISPR platform to screen for human genes that modulate ACE2 expression. ACE2 is the cellular receptor for a few human coronaviruses, i.e. SARS-CoV-1, SARS-CoV2, and HCoV-NL63. The authors hypothesized that some of the previously reported SARS-CoV-2 host factors (validated or not) might be involved in regulating ACE2 (SARS-CoV-2 receptor) expression. They have selected ~800 of these host factors as targets for engineering a custom-designed CRISPR KO guide RNA library. To screen for host genes that could potentially regulate ACE2 expression, the authors choose a human hepatoma cancer cell line, Huh7, that is known to be a susceptible & permissive cell line to SARS-CoV-2. Two different populations of Huh7 cells were used – an original population that consists of only a small % of cells expresses ACE2 and an ACE2-enriched population (pre-sorted by flow cytometry). These two different populations of Huh7 cells were mutagenized by the custom CRISPR library, allowed to proliferate, and later sorted by flow cytometry to harvest ACE2 positive/negative cells. Using NGS and bioinformatics, the authors identify 35 candidate genes from the screen and subjected 20/35 of these genes for validation (i.e. measuring % of ACE2 positive cells present in each genetically perturbed cell population). Most of these genes (but not all) were validated as ACE2 levels seemed to be altered upon CRISPR KO in Huh7 cells (both original population and ACE-enriched population). Using pathway analyses – enrichment of GO terms (Molecular Functions) and STRING, the authors suggested that some of the validated hits and some yet-to-be validated hits from the screen may transcriptionally regulate ACE2 expression in the Huh7 cells. Compared to published datasets, the authors propose establishing a correlation between genetic perturbations of a few validated and yet-to-be validated hits on this screen, and resistant phenotypes against SARS-CoV-2 and HCoV-NL63. The authors further examine the ACE2 protein levels in a few CRISPR KO Huh7 cell populations (not sure if this was the original or ACE2-enriched population) by immunoblotting and immunofluorescent imaging, as well as determine the % of ACE2 positive cells by flow cytometry. ACE2 protein/mRNA levels seemed to be dramatically reduced upon depletion of SMAD4, EP300, and PIAS1. In contrast, more abundant ACE2 protein/mRNA was observed in BAMBI KO cells. Subsequently, they demonstrate SARS-CoV-2 infection and propagation in all of these KOs (again, not sure if these derived from the original Huh7 or ACE2-enriched Huh7), except BAMBI, were significantly reduced. BAMBI KO, in contrast, promoted SARS-CoV-2 infection and propagation.

Overall the manuscript is well written, but some hiccups are spotted. For example, a few supplementary figures are either provided without being mentioned in the text (e.g. Fig S2, 3E-3H & S4) or inaccurately cited in the text (e.g. Page 8: “We observed that gene disruptions which reduced LDL uptake were also more likely to confer cellular resistance to SARS-CoV-2 both at 37⁰C and at 33⁰C (Figure S3A).” Please check the first paragraph of Page 9, too).

Below are specific comments:

1. The authors cited Wei Jin et al. (2021, Cell) in the manuscript without mentioning a key finding in the article – HMGB1 as a novel regulator of ACE2 expression and a critical factor for SARS-CoV-1, SARS-CoV-2, and HCoV-NL63 infections. Please explain the reason(s) for omitting this important discovery, which is perfectly relevant to this manuscript.

2. Validated genes (i.e. 18/20) – candidate genes considered as validated ACE2 regulators were not clearly defined in the text. Two sets of validations were presented (Fig. 2A and 2B), >2 genes from these validations were clearly shown to give negligible impact to influence % of ACE2 expressing cells upon knockout in original Huh7 and ACE2-enriched population. For instance, % of ACE2 positive for HNF4A KO, BRD2 KO, and ARNT KO in the original Huh7 look the same as the WT/NT. For the ACE2-enriched population, BRD2 KO, TFAP4 KO, KMT2D KO, and ARNT KO did not seem to alter % of ACE2 expressing cells. Please clarify, and if possible, please provide statistical evaluations for data presented in Fig. 2A and 2B.

3. A significant number (9/20) of the top hits selected for validation in Fig. 2 has been previously identified as “strongly selective cancer dependency genes” (DepMap definition: “whose dependency is at least 100 times more likely to have been sampled from a skewed distribution than a normal distribution”) from a huge collection of CRISPR screens for cancer cell lines. For example,

1) PIAS1 (https://depmap.org/portal/gene/PIAS1?tab=overview&characterization=expression)

2) HNF4A (https://depmap.org/portal/gene/HNF4A?tab=overview&characterization=expression)

3) BAMBI (https://depmap.org/portal/gene/BAMBI?tab=overview&characterization=expression)

4) KAT6A (https://depmap.org/portal/gene/KAT6A?tab=overview&characterization=expression)

5) TFAP4 (https://depmap.org/portal/gene/TFAP4?tab=overview&characterization=expression)

6) NR6A1 (https://depmap.org/portal/gene/NR6A1?tab=overview&characterization=expression)

7) GREB1L (https://depmap.org/portal/gene/GREB1L?tab=overview)

8) ZNF217 (https://depmap.org/portal/gene/ZNF217?tab=overview)

9) HHEX (https://depmap.org/portal/gene/HHEX?tab=overview&characterization=expression)

As mentioned in the Materials and Methods, during the CRISPR screen for identifying ACE2 regulators, populations of mutagenized Huh7 cells (cancer cells) were cultured and allowed to proliferate for 14 days. Would it be possible that any of these “strongly selective” cancer dependency genes confound the interpretation on the CRISPR screen hits? Please discuss.

4. STRING analysis – why was ACE2 excluded from the analysis? What criteria were set for the analysis, e.g. network type, meaning of network (evidence/confidence), active interaction sources (text-mining/experiments/databases/Gene Fusion/etc.)? What was the rationale for including invalidated hits (HNF4A, BRD2, TFAP4, and ARNT) in this analysis (and in Fig. 3A, 3C and 3D)?

5. Promising results were demonstrated for mechanistic dissections of SMAD4, EP300, PIAS1, and BAMBI in regulating ACE2 expression, as well as SARS-CoV-2 infection. However, all of these efforts were performed based on the same cancer cell line, in fact, only a single cell line was used throughout the entire manuscript. In addition, these genes are depleted using a single guide RNA, without any cDNA rescue/addback to cross-validate and rule out any off-target effects. It would be crucial to strengthen these important findings by using additional cell lines/human primary cells, and additional guide RNA(s). All CRISPR KO Huh7 cells reported in this manuscript were generated using a lentivirus system, which involved random integration of Cas9 and guide RNAs into the Huh7 genome. Would that lead to an off-target impact? Would it be possible to transiently introduce Cas9 and guide RNAs into cells (without integration) to knockout these genes for cross-checking the impact on ACE2 expression?

6. It is reassuring to see SARS-CoV-2 infection, production, and spread was blocked/promoted in the KO cells at 48 hpi, after multiple rounds of infection. However, the role of these genes in regulating ACE2-mediated entry of the virus was not specifically tested. This key point could perhaps be addressed by standard virus entry assays using either a surrogate system (BSL2) such as recombinant VSV expressing Spike protein of SARS-CoV-2, or HCoV-NL63 (ACE2 dependent, BSL2 pathogen). No additional BSL3 work is required.

Minor comments:

1. Please label the protein standard for all western blots.

2. Page 6: “In accordance with the depth of library coverage in this screen, we found >99.9% library representation in each sorted cell population with minimal skewing of gRNA representation.”

Is “>99.9%” referred to the entire library against “833 selected genes”? This statement is pretty confusing as each sorted cell population should theoretically consist of a distinctive gene pool since each group exhibited different levels of ACE2 expression (i.e. ACE2 negative or positive). Perhaps I simply misunderstood the overall design of the screen and/or the statement.

2. Pg. 18 “Immunobloting of Huh7 RIPA lysates……. against….. SMAD4, EP300, PIAS1” data not shown in the results. Please either share the results or revise the Materials & Methods. Totally understand that the authors may have gone through multiple rounds of manuscript prep; please check through again as multiple hiccups have been spotted.

Reviewer #2: In their manuscript, “Identification of ACE2 modifiers by CRISPR screening,” Sherman et al. aim to identify genes that modify ACE2 surface expression in human hepatocarcinoma cells. SARS-CoV-2, the etiological agent of COVID-19, requires surface expression of ACE2 for cell entry and for productive infection, making modulation of ACE2 expression an enticing therapeutic target. In the hepatoma cells investigated here, as in other human tissues, ACE2 expression levels are heterogenous, and the genetic/epigenetic regulators of these expression patterns are incompletely defined. By employing a set of CRISPR Loss-of-Function screens, the authors identify 35 candidate genes that potentially modulate ACE2 surface expression. In subsequent validation experiments, the authors validate the effects of a subset of screen hits and implicate central mediators of the SMAD4 signaling axis in regulating ACE2 expression. Finally, CRISPR-mediated gene deletion of several top hits is employed to demonstrate functional consequences on the ability of SARS-CoV-2 to productively infect HUH-7 cells.

The studies detailed in the manuscript address a significant research question, with implications for potential therapeutic strategies. The rationale for focused CRISPR screens is clearly explained, and analyses are appropriately implemented. Validation experiments are properly controlled and confidently ascribe ACE2 expression regulatory effects to multiple screen hits. Previously published data is used extensively to correlate ACE2 screen hits with the effects of gene deletion on SARS-CoV-2 infection (Schneider et al., 2021), and LDL uptake (Emmer et al., 2021). However, despite convincing identification and validation of cellular factors that modulate ACE2 expression, there are significant concerns regarding the generalizability of the findings. Moreover, while comparisons to published screens provide important context for the current work, attempts to directly correlate per gene effects may not be fully appropriate due to differences in cell lines and experimental systems. While the SMAD4 signaling axis appears to play a significant role in modulating ACE2 expression, elucidating underlying mechanisms would require additional experimentation. Finally, the manuscript is missing numerous important methodological details and figure annotations. Specific major and minor points are detailed below.

**Part II – Major Issues: Key Experiments Required for Acceptance**

Reviewer #1: Please see Part I.

Reviewer #2: Cell line used and generalizability of results: The Huh7 hepatoma cell line seems to recapitulate the sporadic nature of ACE2 expression in human tissues, making it a good choice for the CRISPR screens described here. In addition, the “two pronged” screen approach (using “non-enriched” and ACE2 expression-enriched populations) effectively improves robustness and hit detection. However, despite comparisons to published data, there is limited evidence provided that the described hits are likely to modulate ACE2 surface expression in additional (and perhaps more physiologically relevant for SARS-CoV-2 infection) cell lines. Additional gene deletion experiments for top screen hits in other cell types are needed to evaluate the generalizability of these effects. Potentially informative cell types include line with high ACE2 expression (e.g., HepG2 cells) and low ACE2 expression (e.g., A549), or perhaps primary airway epithelial cells (as a physiological target of SARS-CoV-2 infection – although this may be beyond the scope of the present manuscript).

In addition, interpretation of screen results would benefit from additional characterization of the Huh7 cell lines used. Specifically, the stability of ACE2 expression in serially-enriched ACE2 cells should be assessed. Given that the guide enrichment/depletion is evaluated in populations FACSorted based on ACE2 expression, “baseline” instability of ACE2 expression, particularly in the enriched group, might impact screen results.

Proposed mechanisms of ACE2 regulation: Screen results and validation experiments implicate SMAD4 signaling components in regulation of ACE2 surface expression. Although the manuscript includes studies demonstrating that these effects operate at the level of mRNA transcription, the underlying mechanistic details are not explored. With the convincing results for SMAD4, additional investigation of TGFbeta/BMP signaling pathways could be investigated for effects on ACE2 expression levels, with different pathway agonists that activate different SMAD complexes evaluated for effects on ACE2 transcription. Whether or not the effects of SMAD4 are direct or indirect could be explored with analyses of the ACE2 promoter and/or SMAD4 binding.

Given the diverse roles of SMAD4 (and other focused hits) on cell biology, it would also be of interest to assess how SMAD4 disruption (and disruption of additional related screen hits) affects transcription more generally (e.g. through RNA-Seq comparisons), that might also point towards indirect effects on ACE2 expression and/or SARS-CoV-2 infection efficiency.

Confirmation of CRISPR-mediated gene disruption: Experimental design and well controlled experiments strongly suggest that disruption of multiple major screen hits alters ACE2 surface expression levels. However, with the exception of ACE2, there are no data presented that validate the efficiency of CRISPR-mediated gene deletion. The efficiency of gene depletion should be confirmed for focused hits, and ideally presented alongside with corresponding ACE2 expression.

**Part III – Minor Issues: Editorial and Data Presentation Modifications**

Reviewer #1: Please see Part I.

Reviewer #2: • While a schematic of screen design and interpretation is helpful in understanding the work, as presently constructed, Figure 1A does not clearly portray the screening methodology.

• Figure S2 C-D, Figure S3 C-D: QC results should include log2 fold-change values for negative controls in the guide library.

• Both whole genome and focused CRISPR screens were properly conducted in multiple biological replicates. However, how these replicates were pooled and analyzed is not described. Relatedly, while the Methods section contains details on the initial whole genome screen, it appears that a description of the focused screen has been omitted.

• CRSIPR screen supplementary tables should include a legends and explanations for the different columns

• Validation experiments, Figure 2, S6, S7: Results as presented indicate the degree to which gene deletion impacts the proportion of ACE2-posiitve cells. It would be informative to also assess and present how deletion affects the abundance of ACE2 on cells (e.g. by mean fluorescence intensity).

• Figure 2 does not report statistical tests. As a result, it is unclear which genetic knockouts result in statistically significant changes to ACE2 expression. In addition, in ACE2-enriched cells, data points for replicate 1 and replicate 3 appear to be very similar, while this is not the case for replicate 1 and replicate 3 measurements in non-enriched Huh7 cells. It should be made clear if these are technical or biological replicates.

• Representative flow cytometry gating schemes should be included as supplemental figures. Moreover, ACE2 flow cytometry gates were set based on unlabeled cells; Given the many problems associated with ACE2 antibody surface labeling previously reported by the authors, was there an isotype control included for setting gates and/or assessing non-specific labeling?

• For experiments presented in Figure 2, it is not clear if the cells were permeabilized prior to staining. The preprint cited in the text does include the methodological details.

• The comparison of screen results to Schneider et al screen data presented in Figure 3C is missing several important annotations and methodological details that make interpretation challenging. It is unclear what values are plotted on heatmaps in Figure 3C – the color scale should be marked accordingly. Moreover, the exact source of values from Schneider et al. and how the comparisons were conducted should be reported.

• References to figure S4 are missing from the text.

• Figure S4A: what genes are considered to be “other genes”?

• Figure 4B should include statistical testing. Furthermore, additional clarification on the complementary methods used for figures 4B-D should be provided in the corresponding text.

• Micrographs in Figure 4C should include a color key and scale bars. Furthermore, as only 3-5% of these cells (controls) are expected to express ACE2 (as reported), an analysis of multiple fields and methodological details on how they were selected should be included.

• In Figure 4D, it is unclear why the y-axis contains a “break” between identical values – perhaps a log10 scale without a break would be equivalent? In addition, the corresponding Methods section should indicate how these values were calculated, and additional details on the panel of control genes used normalization.

• Figure 5A describes the functional outcome of ACE2 reduction after CRISPR gene disruption on SARS-CoV-2 infection. While an M.O.I. of 48 was used for infection, the percentage of infected cells at 48 hours is unexpectedly low. This may be due to virus titration on a more permissive cell line (i.e. Vero-E6).

• The data presented in Figure 5A were collected by high-throughput microscopy on 384 well plates. Specific methodological details, including how images were obtained, how many fields were examined, magnification settings, and the analysis workflow used to calculate infection percentages should be provided.

• In general, the Methods section should be significantly expanded, with additional details for each experimental approach and analysis presented.

PLOS authors have the option to publish the peer review history of their article (what does this mean?). If published, this will include your full peer review and any attached files.

Reviewer #1: No

Reviewer #2: No
---

## [Decision Letter · Decision Letter 1]

2 Feb 2022

Dear Dr. Emmer,

Thank you very much for submitting your manuscript "Identification of cell type specific ACE2 modifiers by CRISPR screening" for consideration at PLOS Pathogens. As with all papers reviewed by the journal, your manuscript was reviewed by members of the editorial board and by several independent reviewers. The reviewers appreciated the attention to an important topic. Based on the reviews, we are likely to accept this manuscript for publication, providing that you modify the manuscript according to the review recommendations.

All of the reviews should be able to be addressed by text edits and discussion.

Sincerely,

Stacy M Horner

Associate Editor

PLOS Pathogens

Andrew Pekosz

Section Editor

PLOS Pathogens

Kasturi Haldar

Editor-in-Chief

PLOS Pathogens

orcid.org/0000-0001-5065-158X

Michael Malim

Editor-in-Chief

PLOS Pathogens

orcid.org/0000-0002-7699-2064

Reviewer Comments (if any, and for reference):

Reviewer's Responses to Questions

**Part I - Summary**

Reviewer #1: Sherman and co-workers have addressed almost all reviewers’ comments and revised the manuscript. The key message for the revised version is that these ACE2 modifiers are likely to be cell type specific, as demonstrated by a new screen using an additional cell line (Calu-3). The authors have clearly stated the limitation of the findings in the revised manuscript.

Minor comments:

Fig. 2A and 2B

Good to see that statistical evaluations have been included. However, the data points plotted for each gene look different from the original submission. Also, data points for WT are removed. Is the original dataset used for these panels replaced by a newer version? Please clarify.

Fig 3

As shown in Fig 2, BRD2, MIDN, and ARNT (the 3/20 invalidated hits) do not impact ACE2 expression in any cell lines. It is puzzling to see these genes appearing in Fig 3 as “validated modifiers” of ACE2. Would it make more sense to exclude these invalidated hits in Fig 3? If not, please justify.

Fig 5

A standard practice to rule out off-target effects is to complement KO cells with respective cDNAs and see if ACE2 expression/virus infectivity can be restored. Perhaps it would be challenging to fine-tune the expression of these ACE2 modifiers ectopically. Have the authors at least looked into that? If not, please justify and state the limitation (i.e., potential off-target effects).

Reviewer #2: In their revised manuscript, “Identification of cell type specific ACE2 modifiers by CRISPR screening” Sherman et al. expand their efforts to elucidate the regulatory pathways that control the expression of ACE2 in human cell lines. The updated submission incorporates several new experiments, most in direct response to the prior reviews. Perhaps the most significant issue in the initial submission, noted by both reviewers, was the investigation of a single cell line (Huh7) for the identification of factors that modulate ACE2 expression (with notably heterogeneous expression patterns), which raised concerns about the generalizability of the findings. In response, the authors now report additional focused CRISPR screens for ACE2-modulating genes in Calu-3 cells. These screens, and the complementary validation experiments that accompany them, are well-designed, conducted with appropriate controls, and yield robust, biologically-interesting results. The Calu-3 lung adenocarcinoma cell line is a good choice for these studies and has been a workhorse cell line for multiple SARS-CoV-2 studies. The authors demonstrate that ACE2 expression patterns in Calu-3 are similarly heterogeneous to Huh7. However, remarkably, the genes identified as modulating ACE2 expression in Calu-3 cells were distinct from those identified in Huh7 cells. Thus, at least for the two cell lines evaluated, the screens were not able to identify “central” and/or “common” regulators of ACE2. The authors note that this outcome highlights the cell type-specific nature of ACE2 expression regulation, which in turn could impact cell type-specific phenomena related to SARS-CoV-2 infection. These cell type-specific results could represent significant and biologically relevant findings; as such, they likely warrant comparative studies exploring the distinct ACE2 regulatory mechanisms at play in different cell types, which are absent from the revised manuscript. The authors might explore genetic, epigenetic, and/or signaling pathway differences in Calu-3 and Huh7, much of which could be assessed using publicly available datasets. For example, the apparent absence of SMAD4-mediated regulation of ACE2 expression in Calu-3 cells (as compared to Huh7 results) is intriguing, and the expression of SMAD4 pathway components could be measured. In summary, in responding to initial reviews, the authors added well-conducted experiments with intriguing results, that unexpectedly revealed cell-type specific ACE2 regulations. Some additional indication as to the underlying reason(s) for the discrepancy between the tested cell lines would dramatically increase impact and context of the study.

**Part II – Major Issues: Key Experiments Required for Acceptance**

Reviewer #1: (No Response)

Reviewer #2: See summary above.

**Part III – Minor Issues: Editorial and Data Presentation Modifications**

Reviewer #1: (No Response)

Reviewer #2: Additional points:

* The authors have appropriately updated the data and text throughout the manuscript as suggested in the previous reviews

* In comparing their Huh7 screen results to the Schneider et al 2021 Huh7.5 SARS-CoV-2 results, it is important to acknowledge the biological differences of Huh7.5 cells (which are dysfunctional in viral RNA sensing) relative to Huh7 cells, particularly when interpreting screen results based on virus cytopathic effects. In addition, the source of the data (Schneider et al) should be noted on the figure depicting the comparisons.

PLOS authors have the option to publish the peer review history of their article (what does this mean?). If published, this will include your full peer review and any attached files.

Reviewer #1: No

Reviewer #2: No

Figure Files:

Data Requirements:

Reproducibility:

References:

---

## [Editor Report · Decision Letter 2]

15 Feb 2022

Dear Dr. Emmer,

We are pleased to inform you that your manuscript 'Identification of cell type specific ACE2 modifiers by CRISPR screening' has been provisionally accepted for publication in PLOS Pathogens.

Best regards,

Stacy M Horner

Associate Editor

PLOS Pathogens

Andrew Pekosz

Section Editor

PLOS Pathogens

Kasturi Haldar

Editor-in-Chief

PLOS Pathogens

orcid.org/0000-0001-5065-158X

Michael Malim

Editor-in-Chief

PLOS Pathogens

orcid.org/0000-0002-7699-2064
---

## [Editor Report · Acceptance letter]

24 Feb 2022

Dear Dr. Emmer,

We are delighted to inform you that your manuscript, "Identification of cell type specific ACE2 modifiers by CRISPR screening," has been formally accepted for publication in PLOS Pathogens.

Best regards,

Kasturi Haldar

Editor-in-Chief

PLOS Pathogens

orcid.org/0000-0001-5065-158X

Michael Malim

Editor-in-Chief

PLOS Pathogens

orcid.org/0000-0002-7699-2064